# Deprescribing montelukast in children with asthma: a systematic review

Eleanor Grace Dixon  ,[1,2] Charlotte King,[3] Andrew Lilley,[4] Ian P Sinha,[5] Daniel B Hawcutt[2,6]

[1]Department of Pharmacology and Therapeutics, University of Liverpool Faculty of Health and Life Sciences, Liverpool, UK
[2]Department of Women's and Children's Health, University of Liverpool, Liverpool, UK
[3]Royal Liverpool University Hospital, Liverpool University Hospitals NHS Foundation Trust, Liverpool, UK
[4]Department of Respiratory Medicine, Alder Hey Children's NHS Foundation Trust, Liverpool, UK
[5]Alder Hey Children's NHS Foundation Trust, Liverpool, UK
[6]Alder Hey Children's Hospital Clinical Research Facility, Liverpool, UK

**Correspondence to**
Dr Daniel B Hawcutt;
dhawcutt@liverpool.ac.uk

## ABSTRACT

**Background** National and international asthma guidelines recommend adjusting asthma treatment based on levels of control, yet no guidance is given regarding the stepping-down of montelukast in children and young people (CYP).
**Objective** To systematically review evidence regarding deprescribing montelukast in CYP with established asthma.
**Design** Systematic review.
**Data sources** Embase, Medline, PubMed and CINAHL were searched up to October 2020.
**Study selection** Eligible studies contained patients aged 0–18 years with a diagnosis of asthma, who had been administering montelukast before it was withdrawn. All reasons for withdrawal were included.
**Results** The search identified 197 papers. After deduplication, five papers were included (three randomised control studies and two cohort studies). Four studies observed the impact of montelukast withdrawal for 2 weeks, and one study for 8 weeks. The impact of withdrawal was measured in the studies using a combination of lung tests (eg, forced expiratory volume in 1 s (FEV1), fractional exhaled nitric oxide (FeNO)), asthma scoring methods and exercise challenges. Of the 17 domains in the Core Outcome Set for Clinical Trials in Childhood Asthma, eight outcomes were measured in at least one of the five studies, with all five studies measuring the outcome of 'Lung Function'. No significant differences were found between the montelukast and placebo groups following montelukast withdrawal. Significant differences between the comparator points within the test group were found in nine outcomes across four studies; FEV1/forced vital capacity, FEV1, forced expiratory flows (25%–75%), asthma score (study specific), maximum % fall in FEV1 and time to recovery (post exercise) significantly decreased whereas FEV1/bronchodilator response, FeNO and eNO significantly increased.
**Conclusion** Only limited, contradictory and short-term effects of deprescribing montelukast in CYP with established asthma are presented in literature. Definitive studies determining clinical stability, and impact of deprescribing montelukast in CYP are imperative to improve the safety of asthma treatment in CYP.
**PROSPERO registration number** CRD42020213971.

## INTRODUCTION

Asthma is a disease of lung inflammation and small airway constriction, and affects more than 338 million people globally.[1] It is the most common chronic disease in children

### Strengths and limitations of this study

- ► With montelukast commonly used globally, this review did not limit publication eligibility based on language.
- ► This systematic review was conducted in accordance with Preferred Reporting Items for Systematic Reviews and Meta-Analyses guidelines, and therefore, regarded as methodologically solid by healthcare professionals (audience).
- ► Some papers had to be excluded due to uncertainties regarding patients' formal asthma diagnosis.

and young people (CYP).[2] Asthma can be of variable severity, with symptoms induced by a range of factors such as exercise, viruses or pollen.[3] As a result, the dose and class of asthma medication hugely varies between individuals in order to ensure an effective, personalised treatment plan. This rationale is encouraged by national and international guidelines which aid clinicians in their decision-making.[4–8] However, there are some ambiguities in the current guidelines.

One drug included in most guidelines is montelukast, the 16th most prescribed medication globally in 2020.[9] Although sometimes prescribed as a first-line treatment, montelukast, a leukotriene receptor antagonist, is commonly prescribed as an additional therapy for patients whose asthma is not controlled by inhaled corticosteroids; its use therefore depends on the needs of a patient at a particular time.[7 10] While the addition and stepping down of treatment is encouraged by asthma guidelines, the process of deprescribing montelukast is not clearly described.[4–8] There is clarity about when montelukast is ineffective, where guidelines state that montelukast treatment should be stopped after an initial trial period.[5–7] However, no montelukast-specific guidance is given for the deprescribing of this drug following the achievement of 'good asthma control'—the definition of which is poorly defined.[4] This is a possible consequence of

the lack of data regarding the deprescribing of montelukast in literature.[11]

This systematic review aims to collate the current knowledge base around the deprescribing of montelukast in CYP with asthma. The primary aim was to identify the impact of montelukast withdrawal on paediatric patients' asthma symptoms and control using the Core Outcome Set for Clinical Trials in Childhood Asthma (COS). The longevity in which the impact of montelukast withdrawal was examined for in the literature was also reviewed.

## METHODS

Our systematic review is reported in line with the Preferred Reporting Items for Systematic Reviews and Meta-Analyses guidelines.

### Eligibility criteria

Eligible studies contained patients aged 0–18 years with a diagnosis of asthma and to whom montelukast had been administered before it was withdrawn. Studies which contained both adult (≥18 years) and paediatric data were included if the relevant data (information regarding the deprescribing on montelukast) were recorded separately from the adult data. Human studies in any language and with any publication date were included. All primary research study designs including randomised controlled trials (RCTs) and cohort studies were eligible. Case reports were also included to ensure patient experience regarding the deprescribing of montelukast was included. No narrative reviews or editorials were included. The primary objective was to identify the impact of montelukast withdrawal on a patient's asthma symptoms and control (including lung function, inflammation, etc) using the COS.[12] The secondary objective was to review the time over which the withdrawal of montelukast in CYP was measured.

### Search strategy and study selection

In September 2020, we searched Medline, PubMed, EMBASE and CINAHL using a combination of MeSH and free text subject headings, where appropriate, to include the research question (see online supplemental table S1 for complete search strategy). The primary author (EGD) screened the titles and abstracts of all identified studies using the eligibility criteria. Eligible studies were additionally screened using the full text. This process was repeated independently by the second author (CK) in October 2020. Subsequently, the authors selected the eligible studies; a separate author (DH) independently resolved disagreements between the authors at the full text stage. Reference lists of eligible papers were manually screened for additional papers.

### Patient and public involvement

The paediatric pharmacology team were awarded an NIHR Research Design Service public involvement grant and used this to ascertain input from both the Young Persons Advisory Group as well as a group of young people with asthma based at Alder Hey Hospital about research around the deprescribing of montelukast. The groups were supportive of research into montelukast deprescribing in CYP, and further understanding of this issue.

### Quality assessment

RTCs were appraised using the Cochrane Revised Collaboration's Risk of Bias Tool[13] and cohort studies using The Newcastle-Ottawa Quality Assessment Form[14] (online supplemental tables S2,S3). This was conducted independently by two authors (EGD and DH). There were no discrepancies.

### Data extraction and synthesis

Data were extracted by two authors independently (EGD and DH). From each study, specific data were extracted related to the domains detailed in the COS (including the impact of montelukast withdrawal on asthma exacerbations, lung function and inflammation, and quality of life.[12] Results were additionally categorised into short-term (0–6 weeks), medium term (6 weeks to 6 months) and long-term effects (>6 months).

### Statistical methods

Results are collated and reported descriptively. Meta-analysis was not appropriate.

## RESULTS

After duplicates were removed, the search identified 197 papers. We excluded 182 papers based on the title and abstract and a further 10 following full-text screening. Five papers met the eligibility criteria (figure 1). The eligible papers comprised three RCT and two cohort studies. In

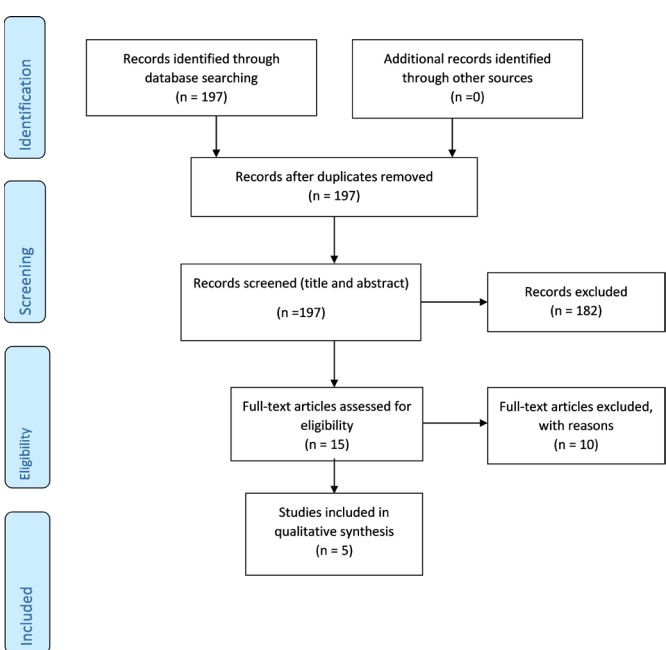

**Figure 1** Search strategy.

total 155 patients were included, of which 107 were both administered, and withdrawn from, montelukast. Further characteristics of the studies are presented in table 1. The design of each study including montelukast administration and withdrawal is shown in figure 2.

## Risk of bias of included studies

Online supplemental tables S2, S3 show the results of the risk of bias assessments for the five papers examined. Both cohort studies show an overall low risk of bias, but one (Lee *et al*) demonstrated some uncertainly due to the selection of the non-exposed cohort (no control group).[15 16] The RCTs were judged to have low risk.[17–19]

## Montelukast withdrawal: comparator points

The studies conducted by Bratton *et al,* Montuschi *et al* and Lee *et al* all measured the impact of montelukast withdrawal using their chosen criteria by comparing asthma symptoms on the final day of montelukast treatment to those on the final day of the wash-out period (figure 2). Montuschi *et al* further compared the treatment group to a placebo group with Bratton *et al* using match controls (no-placebo administered). Lee *et al* did not use a placebo or control group.

Kim *et al* measured the impact of montelukast withdrawal by comparing the asthma symptoms of the 'Period One' treatment group on the final day of montelukast administration (end of week eight) to the asthma symptoms of the 'Period Two' placebo group on the final day of placebo administration (end of week 16). As illustrated in figure 2, at the end of 'Period One', the participants were unblinded, and the placebo group participants were removed from the study. The 'Period One' montelukast group was subsequently re-randomised for 'Period Two'.

Kim *et al* measured the impact of montelukast withdrawal by comparing asthma symptoms at baseline (week 1) with those following placebo administration (week three or week six) (figure 2). It should be noted that montelukast administration for ≥one month was an inclusion criteria in Kim *et al*'s study design and therefore by comparing those participants taking placebo to the baseline, the impact of deprescribing is being measured. The same comparisons were also made following montelukast administration (week 3 or week 6) (figure 2).

## Core outcomes set

Across the five eligible studies, the impact of withdrawing montelukast from children with asthma was only measured against eight of the 17 COS (table 2). A total of 20 outcomes, measured using 13 unique measures, were used to quantify change in individuals before and after montelukast was withdrawn. No data regarding the COS of death, growth, long-term health-related problems, long-term adverse effect, ability to carry out 'normal activities', quality of life, school attendance, general practice/accident and emergency attendance and hospital admission were presented in any of the five studies.

## Exacerbation

Kim *et al*[18] used the Child Asthma Control Test (C-ACT) questionnaire to assess asthma control in both the placebo and test group.[20] No significant difference was found between changes in C-ACT score between treatment groups or the comparator points. Lee *et al*[16] also used an asthma scoring system which monitored factors including asthma exacerbations. Asthma scores did not significantly change during the 2-week wash-out period post montelukast administration (weeks 4–6, figure 2). Full details of this asthma scoring system were not able to be identified. In addition, Lee *et al*[16] recorded that 2 of the 13 patients experienced mild asthma attacks during this 2-week period, demonstrating a worsening of asthma control (table 3).

## Daytime and nocturnal symptoms

The C-ACT questionnaire used by Kim *et al*[18] included an assessment of daytime and nocturnal symptoms.[20] No significant difference was found between changes in C-ACT score between the placebo and test groups or the comparator points.[18] The asthma scoring systems used by Lee *et al*[16] also described monitoring factors including daytime and nocturnal symptoms. The asthma score did not significantly change during the wash-out period (weeks 4–6, figure 2). However, between weeks 8 and 16 (figure 2), there was a significant decrease ($p<0.05$) in the asthma score (improved symptom control) of the period 2 placebo group used by Kim *et al*[17] which accounted for daytime and night-time asthma symptoms (table 4). Full details of this asthma scoring system were not able to be identified.[17]

## Activity or exercise

The C-ACT questionnaire used by Kim *et al* [18] included an assessment of exercise.[20] No significant difference was found between changes in C-ACT score between the placebo and test groups or between the comparator points. However, in the study conducted by Kim *et al*[17] the time taken for a patient's forced expiratory volume in 1 s (FEV1) to return within 10% of their pre-exercise baseline following a standardised exercise challenge significantly decreased ($p<0.050$) between the end of montelukast administration and the end of the wash-out period (weeks 8 and 16 of the period 2 placebo group, figure 2), demonstrating an improvement in asthma control (table 4).

## Short-term adverse effects

Reports of a mild headache subsiding spontaneously 2 days after montelukast withdrawal were recorded by Lee *et al*.[16]

## Lung tests

All five studies undertook lung function tests to assess the impact of montelukast withdrawal on lung function.

The change in FEV1 between comparator points was measured in two of the five studies (table 3).[18 19] Montuschi *et al*[19] recorded a significant decrease ($p=0.011$, table 4) in FEV1 between the final day of montelukast

**Table 1** Eligible studies

| Name of study | Authors | Date | Study type | Study design | Patients who completed the study | Patients given montelukast (n) | Age of patients (years) | Asthma status inclusion criteria | Dose of montelukast (mg) | Other drugs permitted during study |
|---|---|---|---|---|---|---|---|---|---|---|
| Airway mechanics after withdrawal of a leukotriene receptor antagonist in children with mild persistent asthma: Double-blind, randomised, cross-over study | Kim et al[18] | 2020 | Randomised, double-blind, placebo controlled, cross-over study | Placebo or montelukast for 2 weeks, wash-out for 1 week and swap treatment group for 2 weeks. | 28 | 28 | 7.1 (mean) | 1. On Global Initiative for Asthma guidelines step-2 treatment with Montelukast for 1 month or longer. 2. Absence of asthma exacerbations or respiratory symptoms at least 1 month before study onset. 3. No use of systemic corticosteroids during the previous 4 weeks. | 4 or 5 | Short-acting bronchodilator |
| Exhaled nitric oxide (eNO) before and after montelukast sodium therapy in school-age children with chronic asthma: A preliminary study | Bratton et al[15] | 1999 | Cohort study | 2 weeks run-in, treatment for 4 weeks, 2 weeks wash-out. Controls (matched by age and gender, no history of asthma) were given no placebo. | 24 | 12 | 6–11 | 1. Clinical history of mild to moderate stable asthma. 2. Presence of symptoms requiring beta-agonist therapy on at least 7 of the 14 days ruing the run-in period with evidence of airway reversibility (>12% improvement in FEV1). 3. Judged to be in good health on the basis of history and physical examination. | 5 | Constant dose of nasal corticosteroids |
| Effects of montelukast treatment and withdrawal on fractional eNO and lung function in children with asthma | Montuschi et al[19] | 2007 | Randomised, double-blind, placebo controlled, parallel group study | 1-week run-in, treatment or placebo for 4 weeks, 2 weeks wash-out. | 26 | 14 | 10.5 (mean, placebo) 10.8 (mean, treatment) | 1. Have mild persistent asthma defined by the National Heart, Lung and Blood Institute of the National Institutes of Health. 2. Have symptoms more often than twice a week but less often than once a day. 3. FEV1 >80% of predicted value and reversibility of >12% to salbutamol or a positive provocation test result with methacholine or exercise. 4. No leukotriene receptor antagonist in the previous 4 weeks. 5. No glucocorticoids for >4 weeks in previous year. | 5 | Short-acting bronchodilator |
| Effects of montelukast on symptoms and eNO in children with mild to moderate asthma | Lee et al[16] | 2005 | Cohort study | 2 weeks run-in, treatment for 2 weeks, 2 weeks wash-out. | 13 | 13 | 6–14 | 1. Have mild to moderate persistent asthma as defined by the National Asthma Education and Prevention Programme, Expert Panel Report II, Guidelines for the Diagnosis and Management of Asthma. 2. No corticosteroids or leukotriene D4 receptor antagonist treatment in the previous 4 weeks. | 5 | Short-acting bronchodilator |
| Prolonged effect of montelukast in asthmatic children with exercise-induced bronchoconstriction | Kim et al[17] | 2004 | Randomised, parallel-group study | Period 1: Treatment or placebo for 8 weeks (double blind). Period 2: Groups unblinded. Treatment group rerandomised for 8 weeks wash-out or treatment (single blind). | 64 | Period 1: 40 Period 2: 12 | 8–14 (mean, 10.3 treatment) (10.4 mean, placebo) | 1. Have mild asthma defined by the American Thoracic Society. 2. History of exercise-induced bronchoconstriction. 3. FEV1 of >80% predicted and a decrease of >15% after standardised exercise challenge at screening. 4. No corticosteroids, long-acting beta-agonists or leukotriene receptor antagonists. | 5 | Short-acting bronchodilator |

Details of the studies which met the eligibility criteria.
FEV1, forced expiratory volume in 1 s.

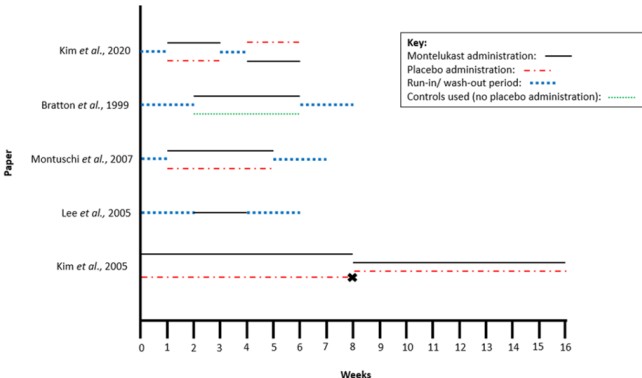

**Figure 2** The design of the eligible studies. Where the participants in studies were split into treatment groups, each group is shown in parallel.

administration and the end of the wash-out period (figure 2) demonstrating a reduced pulmonary function (worsening of asthma symptoms) following montelukast withdrawal (table 4). However, Kim et al[18] recorded no significant change.

Both Kim et al [18] and Montuschi et al[19] recorded a significant decrease (p=0.03 and p<0.003, respectively) in FEV1/forced vital capacity (FVC) values (table 4) between comparator points (figure 2), demonstrating a worsening of asthma symptoms. This was further shown by Kim et al [18] through the significant increase (p=0.04) in FEV1/bronchodilator response between comparator points (figure 2), demonstrating an increase in bronchial

hyper-responsiveness following montelukast withdrawal (table 4). Montuschi et al[19] also described that forced expiratory flows (25%–75%) significantly decreased (p<0.03) between comparator points (figure 2).

Montuschi et al[19] recorded a significant increase (p=0.023) in fractional exhaled nitric oxide (FeNO) following the wash-out phase (weeks 5–7, figure 2), demonstrating an increase in eosinophilic airway inflammation (tables 3 and 4). Changes in eNO recorded by Lee et al[16] also demonstrate a reduced pulmonary function (significant increase, p=0.011) during the wash-out period (figure 2, table 4), however, Bratton et al[15] recorded a non-significant change for the same measure (table 3).

There were no significant changes during the wash-out periods in PEFR and Impulse Oscillometry in Small Airways across any of the five studies (table 3).

### Time frame

Four studies[15 16 18 19] observed the impact of montelukast withdrawal (a wash-out period) for 2 weeks (short term) and one study[17] for 8 weeks (medium term). No study examined the long-term effects (>6 months) of montelukast withdrawal.

### DISCUSSION

Interest in stepping down montelukast has heightened since the FDA applied a boxed warning to montelukast in March 2020 due to the increasing number of neuropsychiatric events reported.[21 22] Since then, further

| Domains of COS | COS for school-aged children with asthma | Kim et al, 2020[18] | Bratton et al, 1999[15] | Montuschi et al, 2007[19] | Lee et al, 2005[16] | Kim et al, 2004[17] |
|---|---|---|---|---|---|---|
| Symptom control | Daytime symptoms | +* | – | – | +* | +* |
| | Death | – | – | – | – | – |
| | Exacerbations | +* | – | – | +* | – |
| | Lung tests | + | + | + | + | + |
| | Nocturnal symptoms | +* | – | – | +* | +* |
| | Parent/child global assessment of control | + | – | – | – | – |
| | Use of reliever inhaler | – | + | – | + | – |
| General health impact | Growth | – | – | – | – | – |
| | Health-related problems when older | – | – | – | – | – |
| | Long-term adverse effect | – | – | – | – | – |
| | Short-term adverse effect | – | – | – | + | + |
| Healthcare required | GP/A+E attendance | – | – | – | – | – |
| | Hospital admission | – | – | – | – | – |
| Life factors | Activity or exercise | +* | – | – | – | + |
| | Normal activities | – | – | – | – | – |
| | Quality of life | – | – | – | – | – |
| | School attendance | – | – | – | – | – |

**Table 2** Presence of core outcome set (COS) in studies

*Parameter included in asthma severity score. Full details of all asthma severity scores were not detailed in the studies.
A+E, accident and emergency; GP, general practice.

**Table 3** Significant differences following the deprescribing of montelukast

| Study | Measure | Significant difference between the test group and placebo following montelukast withdrawal | Significant difference between the test group/ group of interest between the comparator points |
|---|---|---|---|
| Kim et al 2020[18] | FEV1 | No | No |
| | FEV1/FVC | | Yes |
| | IOS | | No |
| | FeNO | | No |
| | C-ACT* | | No |
| | FEV1/BDR | | Yes |
| Bratton et al 1999[15] | eNO | N/A | No |
| Montuschi et al 2007[19] | FeNO | No | Yes |
| | FEV1 | | Yes |
| | FEV1/FVC | | Yes |
| | FEF (25%–75%) | | Yes |
| Lee et al 2005[16] | eNO | N/A | Yes |
| | PEFR | | No |
| | Asthma score | | No |
| | Asthma attack | | N/A (2) |
| Kim et al 2004[17] | Asthma score | No | Yes |
| | Maximum % fall in FEV1 | | Yes |
| | Time to recovery | | Yes |

The recorded outcomes of deprescribing montelukast in children with established asthma.
BDR, bronchodilator response; C-ACT, Child Asthma Control Test; FEF, forced expiratory flows; FeNO, fractional exhaled nitric oxide; FEV1, forced expiratory volume in 1 s; FVC, forced vital capacity; NA, not available.

concerns regarding the adverse drug reactions attributed to montelukast in CYP with asthma have been outlined in a recent PROSPERO systematic review.[23]

Despite these, this review highlights the limited number of studies which have formally examined the stepping down from montelukast in CYP with diagnosed asthma. Within identified studies, only 119 CYP have been included, split between a range of methodologies, all of which limit clinical interpretation and implementation. The five eligible papers primarily examine lung function as a proxy for asthma. Measuring lung function is a commonplace technique used to understand and aid the diagnosis of asthma. However, for most methods of measuring lung function, there is very little information about how the measure relates to asthma management in children in RCT or large longitudinal studies.[24]

For example, Murray et al concluded that the diagnostic algorithm, which uses data from FEV1, FeNO and FVC measurements, was inaccurate.[25] Therefore, the use of lung function measures, of which make up 14 of the 18 measures undertaken across the eligible studies, do not significantly help with the clinical understanding of the impact of montelukast withdrawal in children with asthma.

Additionally, there were few domains from the relevant core outcomes captured and examined. There was only one study that aligned their outcomes with the COS, examining a maximum of 6 of the 17 domains (table 2).[12]

Lastly, the studies examine the effect of the withdrawal of montelukast over a short time frame relative to the expected length of treatment. With four of the five studies only observing the patients for 2-week postfinal montelukast administration, only some aspects of the impact deprescribing montelukast has on patients in the immediate post-administration phase have been examined. As a result, the only knowledge found in literature regarding the effects of deprescribing montelukast for a period greater than 2 weeks comes from 64 paediatric asthma patients in a single study.[17] Further studies examining the deprescribing of montelukast in CYP with asthma over longer time periods to capture events like effects of seasonality, growth and longer-term school attendance and attainment (with regard to known neuropsychiatric adverse effects) are necessary.

### Strengths and limitations of this study

This is the first systematic review of deprescribing montelukast in paediatric asthma patients, and while limited data were identified, this provides a clear direction in terms of the outcomes that need to be captured (improved alignment with the COS) and the study designs (eg, longer time frames) in future research.

However, a limitation of this review is the unknown contents of the two asthma scoring systems used by Lee et al and Kim et al.[16 17] Although some of the factors included in these scores are listed in their publications (eg, nighttime symptoms) and therefore included in table 2, the extent of the scores could not be examined as part of this review. It is possible that certain factors outlined in by the COS for trials of childhood asthma were examined as part of the scoring systems used. However, these were small studies, comprising only 41 paediatric asthma patients in total, and even if a large number of domains with the COS were captured, it is unlikely to have affected our view that the impact on children with asthma is not well described. Additionally, it is possible that individual factors examined in the scoring systems did significantly change between comparator points, but the overall asthma score did not and therefore the change in the factor was masked.

### CONCLUSION

To conclude, the knowledge regarding the impact of deprescribing montelukast in children with asthma

**Table 4** Significant differences in measures between the end of montelukast administration and the end of the wash-out period (figure 2)

| Study | Measure with significant change across studies | Significant change between of test group/ group of Interest | Value at first comparator point | Value at second comparator point |
|---|---|---|---|---|
| Kim *et al* 2020[18] | FEV1/FVC | Significantly decrease (p=0.03) | 86% (95% CI 83% to 89%) | 83% (95% CI 80% to 86%) |
| | FEV1/BDR | Significantly increase (p=0.04) | 6.42 (95% CI 3.74% to 9.11%) | 10.72 (95% CI 5.82% to 15.61%) |
| Montuschi *et al* 2007[17] | FeNO | Significantly increased (p=0.023) | 37.9 ppb (25.6–62.5) (median (IQR)) | 52.2 ppb (36.0–72.9) (median (IQR)) |
| | FEV1 | Significantly decreased (p=0.011) | 2.48 L±0.22 (mean±SEM) | 2.33±0.19 (mean±SEM) |
| | FEV1/FVC | Significantly decreased (p≤0.003) | Not stated in text-graphically presented only | Not stated in text-graphically presented only |
| | FEF (25%–75%) | Significantly decreased (p≤0.03) | Not stated in text-graphically presented only | Not stated in text-graphically presented only |
| Lee *et al* 2005[16] | eNO | Significantly increased (p=0.011) | 13.5 ppb ±7.60 (mean±SD) | 29.2 ppb ±16.1 (mean±SD) |
| Kim *et al* 2004[17] | Asthma score | Significantly decreased (p<0.050) | 17.8±6.8 (mean±SD) | 17.7±6.78 (mean±SD) |
| | Maximum % fall in FEV1 | Significantly decreased (p<0.050) | 27.6±14.4 (mean±SD) | 26.7±19.4 (mean±SD) |
| | Time to recovery | Significantly decreased (p<0.050) | 25.3 mins±23.3 (mean±SD) | 27.7 mins±26.5 (mean±SD) |

BDR, bronchodilator response; FEF, forced expiratory flows; FeNO, fractional exhaled nitric oxide; FEV1, forced expiratory volume in 1 s; FVC, forced vital capacity.

in relation to the COS for Clinical Trials in Childhood Asthma[12] in literature is limited, contradictory and only the short-term effects of stepping down this therapy are known. Definitive studies determining clinical stability, and impact of deprescribing montelukast in CYO are imperative in order for guidelines to fully reflect the overall impact of stepping down this treatment in children.

**Acknowledgements** This is a summary of independent research carried out at the National Institute for Health Research (NIHR) Alder Hey Clinical Research Facility. The views expressed are those of the author(s) and not necessarily those of the NHS, the NIHR or the Department of Health.

**Contributors** EGD conducted the systematic review. This included establishing the eligibility criteria, conducting the search, collecting the data and analysing the results. CK conducted the data search in parallel and compared eligible papers with EGD. AL advised and provided information on deprescribing and prescribing policy and guidelines. IS and DH equally oversaw the direction of the review including editing the writing of the paper and acting as consultants in ensuring clinical relevance of the review.

**Funding** The authors have not declared a specific grant for this research from any funding agency in the public, commercial or not-for-profit sectors. The University of Liverpool funded the open access license.

**Competing interests** None declared.

**Patient consent for publication** Not applicable.

**Ethics approval** This study does not involve human participants.

**Provenance and peer review** Not commissioned; externally peer reviewed.

**Data availability statement** All data relevant to the study are included in the article or uploaded as online supplemental information. The data used in this review was collected from the five eligible studies and therefore available in the public domain.

**ORCID iD**
Eleanor Grace Dixon http://orcid.org/0000-0001-9697-0068

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
