## [Reviewer comments · BMJ Open]

ARTICLE DETAILS

TITLE (PROVISIONAL)	Deprescribing Montelukast in Children with Asthma: A Systematic Review
AUTHORS	Dixon, Eleanor; King, Charlotte; Lilley, Andrew; Sinha, Ian; Hawcutt, Daniel

VERSION 1 – REVIEW

REVIEWER	Désirée Larenas-Linnem Hospital Medica Sur
REVIEW RETURNED	03-Jun-2021

GENERAL COMMENTS	The authors present us here with 5 studies in which MONT was given and then withdrawn in children and young adults with mild asthma, to try to describe the effect on COS, lung function parameters and FeNO of MONT withdrawal. One of the issues that confuses me is that it seems what you would want to compare is parameters of COS at the end of MONT treatment versus XX weeks after its withdrawal (=at the end of 'wash-out'), eventually compared to the parameters at the same time-points in the placebo group (only in the DBPC trials). It is not clear you are really presenting this. You repeatedly refer to 'comparator point', but this is not defined what you mean with this: is it the last day of MONT therapy? Is it the last day of washout, is it pre-montelukast. Please define in detail and then use it consistently throughout the whole text. Moreover, the paper as is needs to undergo thorough grammar check and re-writing as there are several typos and confusing points in the text. Page 2: Abstract results: 'lung function test...' delete function, as FeNO is not a lung function test, but a test reflecting lung eosinophilic inflammation. Thus, one should speak about: lung function tests and FeNO (as the former does not include FeNO). This applies for your whole text and tables. The sentence: '8 of the 17 is very confusing, please restructure. Also the last sentence of results is confusion: ...fall in FEV1....decreased, thus you say FEV1 increased or was improved, I doubt if you really mean that. Asthma score: do you mean asthma symptom score? (ACT or ACQ) What do you mean with recovery time Page 3 Line 7: Strengths: viewed highly... consider change to: regarded as methodologically solid Line 24: I would turn this down: ambiguities on certain aspects in current... Could reduce the quality... (as the recent GLs in general are quite solid in methodology and recommendations, but you are right on some points, specifically stepping down in therapy, there is
--

	little evidence and thus recommendations are uncertain) Lines 37-38: I would say the lack of data on MONT deprescribing in GL is a consequence of the lack to data in clinical trials. There is no solid evidence to base a recommendation on. Consider restructuring this sentence. Line 42: systems???? You mean symptoms Line 43: COS: it is very well possible not all readers know what items COS exactly contains. Please refer to table 2, where you describe COS in detail. Line 44: longevity?? Line 58: 'had been administering...' bad grammar. Change to: to whom montelukast had been administered as maintenance therapy... Page 4: Line 8: again: systems instead of symptoms. FROM HERE ON I SHALL NO FURTHER CORRECT GRAMMAR AND CONCEPT ERRORS. PLEASE HAVE THE PAPER REVIEWED THOROUGHLY AND CORRECTED BY YOUR CO-AUTHORS BEFORE RE-SUBMITTING, as there are many more incorrect issues/expressions Line 8: I have the idea you are also reporting the impact of withdrawal on lung function and FeNO, if so, be complete please. Line 18: 'the remaining studies' ... you mean after pre-selection based on the abstracts? If so, tell so. Line 21: finalized... you mean: selected Line 41: add FeNO In table 1 consider adding outcome measures. Page 8: Line 18: 'factor' probably best called: parameter or item Explanatory asterisks normally go into the footnote of a table, not the header. Line 59: posttreatment is one thing, post washout is another... please clarify what you mean. Table 3: to enhance ease of reading the table, consider fusing all cells in the last column per study and only put once: 'No significant.... Posttreatment in any of these variables.' Page 12: line 11: so the patient was better off after MONT was suspended? Line 24: please specify how much this decline was (% of predicted and if possible mL) and if this is larger than the minimal clinically important difference (MCID, which in adults for VEF1 is 200mL). Line 28: maximum fall.... Do you mean after exercise-challenge or methacholine? Please clarify in the text. Interesting and making one doubt about results: after MONT withdrawal the patient improves.... Line 32: 'reduced pulmonary function'. Please delete, this is not true. FeNO is not a measure of function. I HAVE NOT REVIEWED THE DISCUSSION IN DETAIL YET AS IT HAS TO BE RE-WRITTEN AFTER CORRECTING THE REST. Please add to weaknesses: the very short duration of the trials, thus making it difficult to evaluate the impact of MONT withdrawal in its full length (e.g. on exacerbation frequency and on lung function, for example). Ref 4: please update to GINA 2021
--	--

REVIEWER	Guillermo Sanchez Fundación Universitaria de Ciencias de la salud, SIIES: Research and Education in Health
REVIEW RETURNED	11-Aug-2021

GENERAL COMMENTS	In the abstract the conclusion corresponds to an opinion. The conclusion should be adjusted to the findings of the study. The research question is not clear. I recommend making an explicit statement of it, since it is a relevant point in the systematic review. It appears that the authors propose a systematic prognostic review. It should be clear if it is this type of study. (If yes, I recommend heading to review: https://methods.cochrane.org/prognosis/tools) I recommend clarifying if the objective is: to determine the clinical course of patients with asthma who are suspended from montelukast treatment. The primary and secondary outcomes of interest must be explicit in the methodology. Please indicate why include case reports (it is not highly recommended to mix these studies if other evidence is available). Medline and PubMed are mentioned as consulted bases, explain the difference. EMBASE is mentioned but they only use Mesh terms. Didn't they use Emtree? Clarify if the quality evaluation was in duplicate or by two people, were there discrepancies? How did they solve them? Extraction by a single author can lead to inaccuracies in the data or to incomplete or incorrect information. Why was it not done in duplicate? Why did they use Cochrane's RoB tool, which is now considered obsolete, and not the current version RoB2? Did they plan to do a pooled estimate of the effect and was it not possible or from the protocol they only planted a narrative description of the results? The final conclusion is not clear. It must be based on the evidence reviewed. Specify whether the question is answered or not. State precisely whether first studies are needed to answer the question. For clinicians, what is the message against deprescribing montelukast?
---

REVIEWER	Nadeem Rizvi Jinnah Sindh Medical University, Pulmonology
REVIEW RETURNED	04-Sep-2021

GENERAL COMMENTS	A few typo errors pg 4 line 42 Pg5 line 8
---

VERSION 1 – AUTHOR RESPONSE

Thank you for reviewing my paper. I really appreciate your consideration.

I have made the changes suggested by you and your colleagues. Please thank them for their very useful and detailed feedback.

I look forward to hearing from you.

VERSION 2 – REVIEW

REVIEWER	Guillermo Sanchez Fundación Universitaria de Ciencias de la salud, SIIES: Research and Education in Health
REVIEW RETURNED	18-Oct-2021

GENERAL COMMENTS	Most of the suggestions were included.
--